# BLISS IN NON-ISOMETRIC EMBEDDING SPACES

## ABSTRACT

Recent work on bilingual lexicon induction (BLI) has frequently depended either on aligned bilingual lexicons or on distribution matching, often with an assumption about the isometry of the two spaces. We propose a technique to quantitatively estimate this assumption of the isometry between two embedding spaces and empirically show that this assumption weakens as the languages in question become increasingly etymologically distant. We then propose Bilingual Lexicon Induction with Semi-Supervision (BLISS) — a novel semi-supervised approach that relaxes the isometric assumption while leveraging both limited aligned bilingual lexicons and a larger set of unaligned word embeddings, as well as a novel hubness filtering technique. Our proposed method improves over strong baselines for 11 of 14 language pairs on the MUSE dataset, particularly for languages whose embedding spaces do not appear to be isometric. In addition, we also show that adding supervision stabilizes the learning procedure, and is effective even with minimal supervision.

## 1 INTRODUCTION

Bilingual lexicon induction (BLI), the task of finding corresponding words in two languages from comparable corpora (Haghighi et al., 2008; Xing et al., 2015; Zhang et al., 2017b; Artetxe et al., 2017; Lample et al., 2018), finds use in numerous NLP tasks like POS tagging (Zhang et al., 2016), parsing (Xiao & Guo, 2014), document classification (Klementiev et al., 2012), and machine translation (Irvine & Callison-Burch, 2013; Qi et al., 2018).

Most work on BLI uses methods that learn a mapping between two word embedding spaces (Ruder, 2017), which makes it possible to leverage pre-trained embeddings learned on large monolingual corpora. A commonly used method for BLI, which also empirically works well, involves learning an orthogonal mapping between the two embedding spaces (Mikolov et al. (2013a), Xing et al. (2015), Artetxe et al. (2016), Smith et al. (2017)). However, learning an orthogonal mapping inherently assumes that the embedding spaces for the two languages are isometric (subsequently referred to as the orthogonality assumption). This is a particularly strong assumption that may not necessarily hold true, and consequently we can expect methods relying on this assumption to provide sub-optimal results. In this work, we examine this assumption, identify where it breaks down, and propose a method to alleviate this problem.

We first formally describe this orthogonality assumption, and then present a theoretically motivated approach based on the Gromov-Hausdroff (GH) distance to check the validity and extent of the orthogonality assumption (§2). We show that the constraint indeed does not hold, particularly for etymologically distant language pairs.

SpeMotivated by the above observation, we propose a framework for **B**ilingual **L**exicon **I**nduction with **S**emi-**S**upervision (**BLISS**) (§3.2) that alleviates the aforementioned issues. Moreover, besides addressing the limitations of the orthogonality assumption, the semi-supervised framework also addresses the shortcomings of purely supervised and purely unsupervised methods for BLI (§3.1). Our framework jointly optimizes for supervised embedding alignment, unsupervised distribution matching, and a weak orthogonality constraint in the form of a back-translation loss. Our results show that the different losses work in tandem to learn a better mapping than any one could on its own (§4.3). We also show that the proposed framework improves performance over strong baselines on two datasets, particularly for the case of embedding spaces where the orthogonal assumption is not valid.

Our analysis (§4.3) demonstrates that adding supervision to the learning objective, even in the form of a small seed dictionary, significantly improves the stability of the learning procedure. In particular, for cases where either the embedding spaces are far apart according to GH distance or the quality of the original embeddings is poor, our framework converges where the unsupervised baselines fail to. We also show that for the same amount of available supervised data, leveraging unsupervised learning allows us to obtain superior performance over baseline supervised and unsupervised methods using a comparable amount of data. All our code can be found at `www.toaddhereifaccepted.com`.

## 2 ISOMETRY OF EMBEDDING SPACES

Both supervised and unsupervised BLI often rely on the assumption that the word embedding spaces are isometric to each other. Thus, they learn an orthogonal mapping matrix to map one space to another. For example, for the case of supervised Bilingual Lexicon Induction, Xing et al. (2015) learn an orthogonal mapping matrix to minimize the distance between the projected source and the target embeddings; while in the unsupervised case, Lample et al. (2018) propose learning a matrix near the manifold of orthogonal matrices to match the distributions of the projected source and target word embeddings.

We hypothesize that this assumption might not always hold, in particular for the cases when the language pairs in consideration are etymologically distant — Zhang et al. (2017a) and Søgaard et al. (2018) provide evidence of this by observing a higher Earth Mover's distance and eigenvector similarity metric respectively between etymologically distant languages. In order to test this hypothesis, we propose a novel way of a-priori analyzing the validity of the orthogonality assumption using the Gromov Hausdorff (also referred as GH) distance.

In order to analyze the validity of the orthogonality assumption, we quantitatively check how well the metric spaces of the word vectors of two languages can be aligned under an isometric transform using the GH distance.[1]

The Hausdorff distance between two metric spaces is a measure of the worst case or the diametric distance between the spaces. Intuitively, it measures the distance between the nearest neighbours that are the farthest apart. Concretely, given two metric spaces $\mathcal{X}$, and $\mathcal{Y}$ with a distance function $d(.,.)$, the Hausdorff distance is defined as:

$$\mathcal{H}(\mathcal{X}, \mathcal{Y}) = \max\{ \sup_{x \in \mathcal{X}} \inf_{y \in \mathcal{Y}} d(x,y), \sup_{y \in \mathcal{Y}} \inf_{x \in \mathcal{X}} d(x,y) \}. \tag{1}$$

The Gromov-Hausdorff distance minimizes the Hausdorff distance over all isometric transforms between $\mathcal{X}$ and $\mathcal{Y}$, thereby providing a quantitative estimate of the isometry of two spaces

$$\mathcal{H}(\mathcal{X}, \mathcal{Y}) = \inf_{f,g} \mathcal{H}(f(\mathcal{X}), g(\mathcal{Y})), \tag{2}$$

where $f, g$ belong to set of isometric transforms.

Computing the Gromov-Hausdorff distance involves solving hard combinatorial problems, but can be tractably approximated using the Bottleneck distance (Chazal et al., 2009). In order to compute the Bottleneck distance between two metric spaces, we compute the first order Vietoris-Rips complex (first order for computational efficiency) at $t$ for both spaces: a graph containing an edge between two points iff they lie within a Euclidean distance $t$ from each other in the metric space. As $t$ is varied, the Vietoris-Rips complex goes from the individual points (at $t = 0$) to a single cluster (at $t = \infty$). As $t$ increases, clusters are formed (birth) and eventually merge together (death). The persistence diagram is a 2D plot of the $(t_{birth}, t_{death})$ of each cluster, where $t_{birth}$ and $t_{death}$ are the values of $t$ at which the cluster was born and died respectively. Given two persistence diagrams $f, g$, let $\gamma$ be a bijective map from the points of $f$ to the points of $g$. The bottleneck distance ($\mathcal{B}$) is then defined as:

$$\mathcal{B}(f, g) = \inf_{\gamma} \left( \sup_{u \in f} ||u - \gamma(u)||_{\infty} \right) \tag{3}$$

---

[1]Note that since we mean center the embeddings, the orthogonal transforms are equivalent to isometric transforms.

| Source → Target | Incorrect Predicted |
|---|---|
| aunt → тетя | бабушка (Grandmother) |
| uruguay → уругвая | аргентины (Argentina) |
| regiments → полков | кавалерийские (Cavalry) |
| comedian → комик | актёр (Actor) |

Table 1: Words for which semi-supervised method predicts correctly, but unsupervised method doesn't. The unsupervised method is able to guess the general family but fails to pinpoint exact match

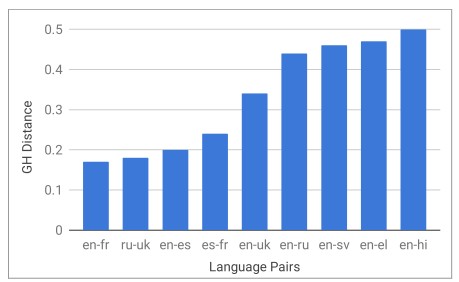

Figure 1: Language Pairs and their GH distance

Chazal et al. (2009) showed that the Gromov-Hausdorff distance can be lower bounded by the Bottleneck Distance between the Persistence Diagrams of the Vietoris-Rips Filtration of the two spaces.

We compute this lower bound for the top frequency words of different language pairs. As can be observed from Figure 1, the GH distances are higher for distant language pairs.

## 3 SEMI-SUPERVISED FRAMEWORK

In this section, we motivate and define our semi-supervised framework for BLI. First we describe issues with purely supervised and unsupervised methods, and then lay the framework for tackling them along with orthogonality constraints.

### 3.1 DRAWBACKS OF PURELY SUPERVISED AND UNSUPERVISED METHODS

Purely supervised methods for aligning word vectors do not utilize the rich information present in the topology of the word vectors. Purely unsupervised methods, on the other hand, can suffer from poor performance if the distribution of the embedding spaces of the two languages are very different from each other. Moreover, unsupervised methods can successfully align clusters of words together, but miss out on fine grained alignment within the clusters.

We explicitly show the aforementioned problem of purely unsupervised methods with the help of the toy dataset shown in 2a, and 2b. In this dataset, due to the density difference between the two large blue clusters, unsupervised matching is consistently able to align them properly, but has trouble aligning the smaller embedded green and red sub-clusters. The correct transformation of the source space is a clockwise 90° rotation followed by reflection along the x-axis. Unsupervised matching converges to this correct transformation only half of the time; in rest of the cases, it ignores the alignment of the sub-clusters and converges to a 90° counter-clockwise transformation as shown in 2c.

We also find evidence of this problem in the real datasets used in our experiments as shown in Table 1. It can be seen that the unsupervised method aligns clusters of similar words, but is poor at the fine grained alignment. We hypothesize that this problem can be resolved by giving it a some supervision in the form of matching anchor points inside these sub-clusters, which correctly aligns them. Analogously, for the task of BLI, generating a small supervised seed lexicon is generally feasible for most language pairs, through either bilingual speakers, existing dictionary resources, or Wikipedia language links. This can provide the requisite supervision.

### 3.2 A SEMI-SUPERVISED FRAMEWORK

In order to alleviate the problems with the orthogonality constraints, the purely unsupervised and supervised approaches, we propose a semi-supervised framework, described below.

Let $\mathcal{X} = \{x_1 \dots x_n\}$ and $\mathcal{Y} = \{y_1 \dots y_m\}$, $x_i, y_i \in \mathbb{R}^d$ be two sets of word embeddings from the source and target language respectively and let $\mathcal{S} = \{(x_1^s, y_1^s) \dots (x_k^s, y_k^s)\}$ denote the bilingual aligned word embeddings. Define $W$ to be the mapping matrix.

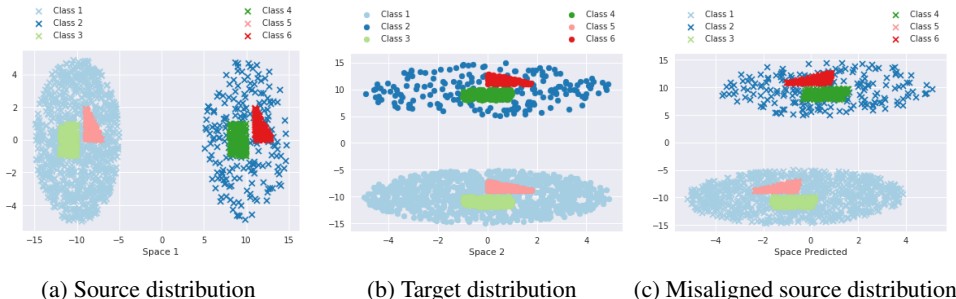

(a) Source distribution      (b) Target distribution      (c) Misaligned source distribution

Figure 2: A toy dataset demonstrating the shortcomings of unsupervised distribution matching. Fig. a) and b) show two different distributions (source and target respectively) over six classes. Classes 1 and 2; classes 3 and 4; classes 5 and 6 were respectively drawn from a uniform distribution over a sphere, rectangle and triangle respectively. Fig. c) shows the misprojected source distribution obtained from unsupervised distribution matching which fails to align with the target distribution of Fig. b).

For learning $W$, we leverage unsupervised distribution matching, aligning known word pairs and a data-driven weak orthogonality constraint.

**Unsupervised Distribution Matching**: Given all word embeddings $\mathcal{X}$ and $\mathcal{Y}$, the unsupervised loss $\mathcal{L}_{W|D}$ aims to match the distribution of both embedding spaces. In particular, for our formulation, we use an adversarial distribution matching objective, similar to the work of Lample et al. (2018). Specifically, a mapping matrix $W$ from the source to the target is learned to fool a discriminator $D$, which is trained to distinguish between the *mapped* source embeddings $W\mathcal{X} = \{Wx_1 \dots Wx_n\}$ and $\mathcal{Y}$. The corresponding objectives are defined as:

$$\mathcal{L}_{D|W} = -\frac{1}{n} \sum_{x_i \in \mathcal{X}} \log(1 - D(Wx_i)) - \frac{1}{m} \sum_{x_i \in \mathcal{Y}} \log D(x_i) \tag{4}$$

$$\mathcal{L}_{W|D} = -\frac{1}{n} \sum_{x_i \in \mathcal{X}} \log D(Wx_i) \tag{5}$$

**Aligning Known Word Pairs**: Given aligned bilingual word embeddings $\mathcal{S}$, we aim to minimize a similarity function ($f_s$) which maximizes the similarity between the corresponding matched pairs of words. Specifically, the loss is defined as:

$$\mathcal{L}_{W|S} = -\frac{1}{|\mathcal{S}|} \sum_{(x_i^s, y_i^s) \in \mathcal{S}} f_s(Wx_i^s, y_i^s) \tag{6}$$

**Weak Orthogonality Constraint**: Given an embedding space $\mathcal{X}$, we define a consistency loss that maximizes the similarity $f_a$ between $x$ and $W^T W x$, $x \in \mathcal{X}$. This cyclic consistency loss $\mathcal{L}_{W|O}$ encourages orthogonality of the $W$ matrix based on the joint optimization:

$$\mathcal{L}_{W|O} = -\frac{1}{|\mathcal{X}|} \sum_{x_i \in \mathcal{X}} f_a(x_i, W^T W x_i) \tag{7}$$

The above loss term allows the model adjust the trade-off between orthogonality for accuracy, based on the joint optimization. This is particularly helpful for cases where the orthogonality constraint is violated, where the embedding spaces are not isometric, specifically for etymologically distant language pairs, as we show in (4.3).

The final loss function for the mapping matrix is:

$$\mathcal{L} = \mathcal{L}_{W|D} + \mathcal{L}_{W|S} + \mathcal{L}_{W|O} \tag{8}$$

$\mathcal{L}_{W|D}$ enables the model to leverage the distributional information available from the two embedding spaces, thereby using all available monolingual data. On the other hand, $\mathcal{L}_{W|S}$ allows for the correct alignment of labeled pairs when available in the form of a small seed dictionary. Finally, $\mathcal{L}_{W|O}$ encourages orthogonality. One can think of $\mathcal{L}_{W|O}$ and $\mathcal{L}_{W|S}$ as working against each other when the spaces are not isometric. Jointly optimizing both helps the model to strike a balance between them in a data driven manner, encouraging orthogonality but still allowing for flexible mapping.

### 3.3 ITERATIVE PROCRUSTES REFINEMENT AND HUBNESS REMOVAL

A common method of improving BLI is iteratively expanding the dictionary and learning the matrix as a post-processing step (Artetxe et al., 2017; Lample et al., 2018). Given a learnt mapping matrix, Procrustes Refinement first finds the pair of points in the two languages that are very closely matched by the mapping matrix and constructs a bilingual dictionary from these pairs. These pair of points are found by considering the nearest neighbors (NN) of the projected source words in the target space. The mapping matrix is then refined by setting it to be the Procrustes solution of the dictionary obtained. Iterative Procrustes Refinement (also referred as Iterative Dictionary Expansion) applies the above step iteratively until some metric (CSLS in case of Lample et al. (2018)) converges.

However, learning an orthogonal linear map in such a way leads to some words (known as hubs) to become nearest neighbors of a majority of other words (Radovanović et al., 2010; Dinu & Baroni, 2014). In order to estimate the hubness of a point, Radovanović et al. (2010) first computed the distribution $N_x(k)$, the counts of all points $y$ such that $x \in k - NN(y)$, normalized over all k. The skewness of this distribution was defined as the hubness of the point, with positive skew representing hubs and negative skew representing isolated points. An approximation to this would be $N_x(1)$, i.e the number of points that x is the 1-NN of.

We use a simple hubness filtering mechanism to filter out words in the target domain that are hubs, i.e., words in the target domain which have more than a threshold number of neighbors in the source domain are not considered in the iterative dictionary expansion. Empirically, this leads to a small boost in performance. In our models, we use iterative Procrustes refinement with hubness filtering at each refinement step.

## 4 EXPERIMENTS AND RESULTS

In this section, we measure the GH distances between embedding spaces of various language pairs, and compute their correlation with several empirical measures of orthogonality. Next, we analyze the performance of the instantiations of our semi-supervised framework for two settings of supervised losses, and show that they outperform their supervised and unsupervised counterparts for a majority of the language pairs. Finally we analyze our performance with varying amounts of supervision and highlight the framework's training stability over unsupervised methods.

### 4.1 GH DISTANCE

To evaluate the lower bound on the GH distance between the two embedding spaces, we select the 5000 most frequent words of the source and target language and compute the Vietoris-Rips complex of the zeroth order (which considers only pairwise edges for identifying clusters). These embeddings are mean centered, unit normed and the Euclidean Distance is used as the distance metric.

Row 1 of Table 2 summarizes the GH distances obtained for different language pairs. We find that etymologically close languages such as en-fr and ru-uk have a very low GH distance and can possibly be aligned well using orthogonal transforms. In contrast, we find that etymologically distant language pairs such as en-ru and en-hi cannot be aligned well using orthogonal transforms.

To further corroborate this, similar to Søgaard et al. (2018) , we compute correlations of the GH distance with the accuracies of several methods for BLI. We find that the GH distance exhibits a

| Prop | ru-uk | en-fr | en-es | es-fr | en-uk | en-ru | en-sv | en-el | en-hi | en-ko | \|Corr\| (GH) | \|Corr\| ($\Lambda$) |
|---|---|---|---|---|---|---|---|---|---|---|---|---|
| GH | 0.18 | 0.17 | 0.2 | 0.24 | 0.34 | 0.44 | 0.46 | 0.47 | 0.5 | 0.92 | * | * |
| $\Lambda$ | 16.4 | 4.1 | 5.9 | 4.1 | 11.7 | 14.7 | 7.3 | 11.5 | 7.7 | 6.6 | * | * |
| MUSE(U) | * | 82.3 | 81.7 | 85.5 | 29.1 | 44.0 | 53.3 | 37.9 | 34.6 | 5.1 | **0.87** | 0.61 |
| RCSLS | * | 83.3 | **84.1** | 87.1 | 38.3 | **57.9** | 61.7 | 47.6 | 37.3 | 37.5 | **0.74** | 0.52 |
| GeoMM | * | 82.1 | 81.4 | **87.8** | 39.1 | 51.3 | **65** | 47.8 | **39.8** | 34.6 | **0.76** | 0.49 |
| BLISS (R) | * | **83.9** | **84.3** | 87.1 | **40.7** | 57.1 | **65.1** | **48.5** | 38.1 | **39.9** | **0.73** | 0.50 |
| $\|\|I - W^T W\|\|^2$ | 0.03 | 0.01 | 0.03 | 0.02 | 59.8 | 54.3 | 71.6 | 72.6 | 106.3 | 98.46 | **0.84** | 0.75 |

Table 2: Correlation of GH and Eigenvector similarity with performance of BLI methods

strong negative correlation with these accuracies, implying that as the GH distance increases, it becomes increasingly difficult to align these language pairs. Søgaard et al. (2018) also proposed the eigenvector similarity metric between embedding spaces for measuring similarity between the embedding spaces. We compute their metric over top $n$ (100, 500, 1000, 5000 and 10000) embeddings (Column $\Lambda$ in Table 2 shows correlation for the best setting of $n$) and show that the GH distance (Column GH) correlates better with the accuracies than eigenvector similarity.

Furthermore, we also compute correlations against an empirical measure of the orthogonality of two embedding spaces by computing $\|\|I - W^T W\|\|^2$, where $W$ is a mapping from one language to the other obtained from an unsupervised method (MUSE(U)). Note that an advantage of this metric is that it can be computed even when the supervised dictionaries are not available (ru-uk in Table 2). We obtain a strong correlation with this metric as well.

## 4.2 Performance of BLISS on Benchmark tasks

### 4.2.1 Baseline Methods

**MUSE (U/S/R)** (Lample et al., 2018) proposed two models: MUSE(U) and MUSE(S) for unsupervised and supervised BLI respectively. MUSE(U) uses a GAN based distribution matching followed by iterative Procrustes refinement. MUSE(S) learns an orthogonal map between the embedding spaces by minimizing the Euclidean distance between the supervised translation pairs. Note that for unit normed embedding spaces, this is equivalent to maximizing the cosine similarity between these pairs. MUSE(R) is the semi-supervised extension of MUSE(S), which uses iterative refinement using the CSLS metric starting from the mapping learnt by MUSE(S). We also use our proposed hubness filtering technique during the iterative refinement process (MUSE(HR)) which leads to small performance improvements. We consequently use the hubness filtering technique in all our models.

**RCSLS** (Joulin et al., 2018) propose optimizing the CSLS metric [2] directly for the supervised matching pairs. This leads to significant improvements over MUSE(S) and achieves state of the art results for a majority of the language pairs at the time of writing.

**VecMap models** (Artetxe et al., 2017) and (Artetxe et al., 2018a) proposed two models, VecMap and VecMap[++] which were based on iterative Procrustes refinement starting from a small seed lexicon based on numeral matching.

### 4.2.2 BLISS models

We instantiate two instances of our framework corresponding to the two supervised losses in the baseline methods mentioned above. BLISS(M) optimizes the cosine distance between supervised matching pairs as its supervised loss ($\mathcal{L}_{W|S}$), while BLISS(R) optimizes the CSLS metric between these matching pairs for its $\mathcal{L}_{W|S}$. Similar to Lample et al. (2018) we use the unsupervised CSLS metric as a stopping criterion during training.

---

[2]Since the CSLS metric requires computing the nearest neighbors over the whole embedding space, this can also be considered a semi-supervised method.

| Model | en-es | es-en | en-fr | fr-en | en-de | de-en | en-ru | ru-en | en-zh | zh-en |
|---|---|---|---|---|---|---|---|---|---|---|
| MUSE (U) | 81.7 | 83.3 | 82.3 | 82.1 | 74.0 | 72.2 | 44.0 | 59.1 | 32.5 | 31.4 |
| MUSE (S) | 81.4 | 82.9 | 81.1 | 82.4 | 73.5 | 72.4 | 51.7 | 63.7 | 42.7 | 36.7 |
| MUSE (R) | 81.9 | 83.5 | 82.1 | 82.4 | 74.3 | 72.7 | 51.7 | 63.7 | 42.7 | 36.7 |
| MUSE (HR) | **82.3** | 83.3 | 82.5 | 83.2 | **75.7** | 72.8 | 52.8 | **64.1** | **42.7** | 36.7 |
| BLISS (M) | 82.3 | **84.3** | **83.3** | **83.9** | 75.7 | 73.8 | 55.7 | 63.7 | 41.1 | **41.4** |
| GeoMM | 81.4 | 85.5 | 82.1 | 84.1 | 74.7 | **76.7** | 51.3 | 67.6 | **49.1** | 45.3 |
| RCSLS | **84.1** | **86.3** | 83.3 | 84.1 | **79.1** | 76.3 | 57.9 | 67.2 | 45.9 | 46.4 |
| BLISS (R) | **84.3** | 86.2 | **83.9** | **84.7** | 79.1 | 76.6 | 57.1 | **67.7** | 48.7 | **47.3** |

Table 3: Performance comparison of BLISS against various baseline models on the MUSE dataset. Numbers in bold correspond to best in the set

| Pairs | # seeds | Vec Map | Vec Map$^{++}$ | MUSE (U) | MUSE (R) | BLISS (M) | RCSLS | BLISS (R) | GeoMM | Vec Map (U)$^{++}$ |
|---|---|---|---|---|---|---|---|---|---|---|
| en-it | all | 39.7 | 45.3 | 45.8 | 45.3 | 45.9 | 45.4 | 46.2 | **48.3** | 48.5 |
|  | Num. | 37.3 | - |  | 0.7 | 44.3 | 0.3 | 44.6 | 1.2 |  |
| en-de | all | 40.9 | 44.1 | 0 | 47.0 | 48.3 | 47.3 | 48.1 | **48.9** | 48.1 |
|  | Num. | 39.6 | - |  | 39.9 | 47.2 | 1.0 | 46.5 | 2.3 |  |

Table 4: Performance of different models on the VecMap dataset

After learning the final mapping matrix, the translations of the words in the source language are mapped to the target space and their nearest neighbors according to the CSLS distance (Lample et al., 2018) are chosen as the translations.

### 4.2.3 DATASETS

We evaluate our models against baselines on two popularly used datasets: the MUSE dataset and the VecMap dataset. The MUSE dataset used by Lample et al. (2018) consists of embeddings trained by Bojanowski et al. (2016) on Wikipedia and bilingual dictionaries generated by internal translation tools used at Facebook. The VecMap dataset introduced by Dinu & Baroni (2014) consists of the CBOW embeddings trained on the WacKy crawling corpora. The bilingual dictionaries were obtained from the Europarl word alignments. We use the standard training and test splits available for for both the datasets.

### 4.2.4 RESULTS

Table 3 shows the performance of BLISS (M) and BLISS (R) against its supervised and unsupervised counterparts on the MUSE datasets. As can be seen, the semi-supervised framework outperforms the both methods for 9 of 10 language pairs for either cases.

Our semi-supervised framework outperforms its supervised and unsupervised counterparts on the VecMap datasets too. Table 4 groups by model category, and contrasts the performance between different models on the VecMap datasets. It can be seen that BLISS(M) and BLISS(R) perform better than the MUSE baselines (MUSE(U), MUSE(R)) and RCSLS respectively.

We also compare against GeoMM (Jawanpuria et al., 2018) and Vecmap $(U)^{++}$ (Artetxe et al., 2018b). These methods learn orthogonal mappings for both source and target spaces to a common embedding space, and subsequently do the translations in the common space. We believe that these models outperform BLISS in the VecMap dataset because BLISS suffers from a slight disadvantage by translating in the target space, as opposed to in the common embedding space. This hypothesis is also supported by the results of Kementchedjhieva et al. (2018).

All the hyperparameters for the experiments can be found in the Appendix (§7.4)

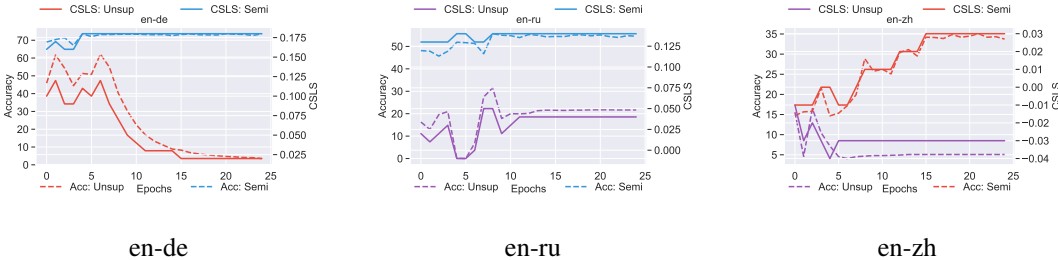

|  | en-de | en-ru | en-zh |
| --- | --- | --- | --- |

Figure 3: Training Stability of different language pairs

### 4.3 Benefits of BLISS

**Languages with high GH distance** BLISS particularly shines over its supervised counterpart when the two embedding spaces are significantly different and the orthogonality constraint is violated. Table 2 shows that BLISS (R) achieves performance gains over RSCLS for language pairs with high GH distance.

**Performance with varying amount of supervision** Table 5 shows the performance of BLISS(R) as a function of the number of data points provided for supervision. As can be observed, the model performs reasonably well even for low amounts of supervision and outperforms the unsupervised baseline MUSE(U) and it's supervised counterpart RCSLS. Moreover, note that the difference is more prominent for en↔zh, whose spaces are not isometric, as can be seen from the GH distance. In this case the baseline models completely fail to train for 50 points, whereas BLISS(R) performs reasonably well. For a detailed ablation, please refer to Appendix Section 8.

**Stability of Training** We also observe that providing even a little bit of supervision helps stabilize the training process, when compared to purely unsupervised distribution matching. We measure the stability during training using

| src-tgt | Model | Num Sup Points | | | |
| --- | --- | --- | --- | --- | --- |
| | | 50 | 500 | 5000 | all |
| en-de | MUSE (U) | | 74.0 | | |
| | MUSE (R) | 31.9 | 73.1 | 75.2 | 75.7 |
| | RCSLS | 0.1 | 9.9 | 72.5 | **79.1** |
| | BLISS (R) | 75.1 | 74.7 | 75.7 | **79.1** |
| de-en | MUSE (U) | | 72.2 | | |
| | MUSE (R) | 72.7 | 72.7 | 72.4 | 72.8 |
| | RCSLS | 0.13 | 10.2 | 70.9 | **76.3** |
| | BLISS (R) | 72.7 | 73.1 | 72.5 | **76.6** |
| en-zh | MUSE (U) | | 32.5 | | |
| | MUSE (R) | 0.3 | 34.5 | 39.2 | 42.7 |
| | RCSLS | 0.0 | 6.6 | 42.5 | 45.9 |
| | BLISS (R) | 32.6 | 36.3 | 42.5 | **48.7** |
| zh-en | MUSE (U) | | 31.4 | | |
| | MUSE (R) | 0.3 | 32.2 | 36.3 | 36.7 |
| | RCSLS | 0.0 | 7.1 | 41.9 | 46.4 |
| | BLISS (R) | 32.5 | 35.1 | 42.8 | **47.3** |

Table 5: Performance with varying Data

both the ground truth accuracy and the unsupervised CSLS metric (which Lample et al. (2018) showed to be correlated with the ground truth accuracy). As can be seen from Figure 3, BLISS(M) is significantly more stable than MUSE(U), converging to better accuracy and CSLS values.

When the word vectors are not rich enough (word2vec (Mikolov et al., 2013b) instead of fastText), the unsupervised method can completely fail to train. This can be observed for the case of en-de in Table 4. BLISS(M) does not face this problem: adding supervision, even in the form of 50 mapped words for the case of en-de, helps it to achieve reasonable performance.

## 5 Related Work

Mikolov et al. (2013a) first used anchor points to align two embedding spaces, leveraging the fact that these spaces exhibit similar structure across languages. Since then, several approaches have been proposed for learning bilingual dictionaries (Faruqui & Dyer, 2014; Zou et al., 2013; Xing

et al., 2015). Xing et al. (2015) showed that adding an orthogonal constraint significantly improves performance, and admits a closed form solution. This was further corroborated by the work of Smith et al. (2017), who showed that in orthogonality was necessary for self-consistency. Artetxe et al. (2016) showed the equivalence between the different methods, and their subsequent work (Artetxe et al., 2018a) analyzed different techniques proposed in various works (like embedding centering, whitening etc.), and showed that leveraging a combination different methods showed significant performance gains.

However, the validity of this orthogonality assumption has of late come into question: Zhang et al. (2017a) found that the Wasserstein distance between distant language pairs was considerably higher , while Søgaard et al. (2018) explored the orthogonality assumption using eigenvector similarity. We find our weak orthogonality constraint (along the lines of (Zhang et al., 2017b)) when used in our semi-supervised framework to be more robust to this.

There has also recently been an increasing focus on generating these bilingual mappings without an aligned bilingual dictionary, i.e., in an unsupervised manner. Zhang et al. (2017b) and Lample et al. (2018) both use adversarial training for aligning two monolingual embedding spaces without any seed lexicon, while Zhang et al. (2017a) used a Wasserstein GAN to achieve this adversarial alignment, and use an earth-mover based fine-tuning approach; while Grave et al. (2018) formulate this as a joint estimation of an orthogonal matrix and a permutation matrix. However, we show that adding a little supervision, which is usually easy to obtain, improves performance. Another vein of research (Jawanpuria et al., 2018; Artetxe et al., 2018b; Kementchedjhieva et al., 2018) has been to learn orthogonal mappings from both the source and the target embedding spaces into a common embedding space and doing the translations in the common embedding space. Artetxe et al. (2017) and Søgaard et al. (2018) motivate the utility of using both the supervised seed dictionaries and, to some extent, the structure of the monolingual embedding spaces. They use iterative Procrustes refinement starting with a small seed dictionary to learn a mapping; but doing may lead to sub-optimal performance for distant language pairs. However, these methods are close to our methods in spirit, and consequently form the baselines for our experiments.

Another avenue of research has been to try and modify the underlying embedding generation algorithms. Cao et al. (2016) modify the CBOW algorithm Mikolov et al. (2013b) by augmenting the CBOW loss to match the first and second order moments from the source and target latent spaces, thereby ensuring the source and target embedding spaces follow the same distribution. Luong et al. (2015), in their work, use the aligned words to jointly learn the embedding spaces of both the source and target language, by trying to predict the context of a word in the other language, given an alignment. An issue with the proposed method is that it requires the retraining of embeddings, and cannot leverage a rich collection of precomputed vectors (like ones provided by Word2Vec (Mikolov et al., 2013b), Glove (Pennington et al., 2014) and FastText (Bojanowski et al., 2016)).

## 6 Conclusions

In this work, we analyze the validity of the orthogonality assumption and show that it breaks for distant language pairs. We motivate the task of semi-supervised BLI by showing the shortcomings of purely supervised and unsupervised approaches. We finally propose a semi-supervised framework which combines the advantages of supervised and unsupervised approaches and uses a joint optimization loss to enforce a weak and flexible orthogonality constraint. We show that our framework obtains gains over several baseline models for numerous language pairs. On analyzing the model errors, we find that a large fraction of them arise due to polysemy and antonymy (An interested reader can find the details in Appendix (§7.2).

An interesting line of future work would be to extend the method proposed here to account for polysemy in translation, possibly by leveraging the work of Upadhyay et al. (2017), which uses multilingual context for sense disambiguation. Another confounding factor is synonyms and antonyms, which appear in similar contexts, and often incorrectly get translated to each other: leveraging the work done by Mrkšić et al. (2016) and Faruqui et al. (2014) might be an interesting way to mitigate this problem.

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

# 7 APPENDIX

## 7.1 TOY DATASET EXPERIMENT

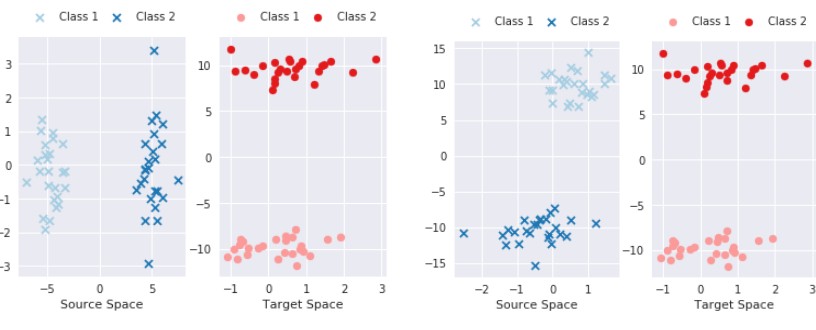

(a) Actual distributions      (b) Misaligned source and actual target distribution

Unsupervised distribution matching solution remains invariant to the permutations of the words within a language. One can easily construct a toy dataset on which this scenario can arise. 4a shows a source and target distribution. Although, in the real dataset each point carries a different label, we only consider 2 labels for the sake of simplicity. The correct transformation for matching distribution *and* labels is an anticlockwise rotation on the source. Since the GAN does not see the labels, it just matches the distributions by half of the times either choosing the correct anticlockwise rotation or the incorrect clockwise rotation on source 4b. This problem can be solved by giving some labeled data correspondence and adding a supervised loss term.

## 7.2 ANALYZING MODEL ERRORS

We characterize the mistakes made by the model, and find that most fall into the following 4 categories:

**Polysemy on the target side**: These are the cases in which the predicted words and the gold translation are synonyms/hypernyms/hyponyms of each other.

**Polysemy on the source side**: These are the cases in which the gold translations and the predicted words are *different senses* of the source word.

**Antonyms**: The distribution of the context of antonyms is often very similar. Unsurprisingly the word vectors of antonyms are quite similar. This leads to cases where the predicted words and gold labels are antonyms of each other.

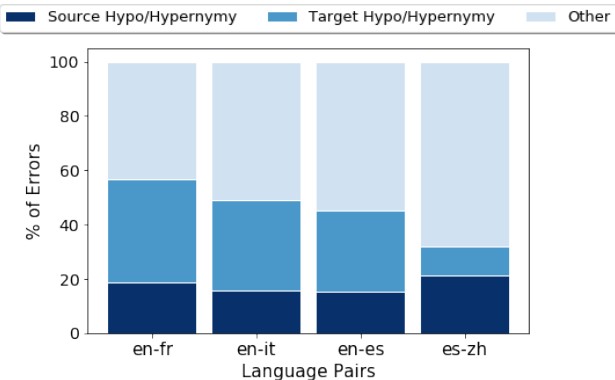

Figure 4: Fraction of errors coming from polysemy in the source/target side and antonymy, for the language pairs en-zh, en-it, en-es and en-fr

**Words that occur in common contexts**: Words that occur in numerous contexts often have poor word embeddings, since a single embedding can't capture polysemy. Consequently, multiple such word embeddings that are frequent and have poor representations often get incorrectly translated to each other. Some examples include proper nouns and numbers

We quantitatively estimate the fraction of errors due to these reasons using WordNet synsets. Given 2 synsets, WordNet provides a score denoting how similar two word senses are, based on the shortest path that connects the senses in the is-a (hypernym/hypnoym) taxonomy. The score is in the range 0 to 1. A score of 1 represents identity i.e. comparing a sense with itself will return 1.

We approximate the fraction of target polysemy errors by finding those cases for which the aforementioned similarity scores between the synsets of the predicted words and the gold translations $\geq 0.1$. Similarly we approximate the fraction of source polysemy errors by finding those cases for which the similarity scores between the synsets of the source word and the predicted word $\geq 0.1$. Fig 4 shows these estimations for different language pairs. See Table 5a in (§7.2) for examples sampled from each of these error types.

| Type of Error | Source | Gold | Predicted | Comments |
|---|---|---|---|---|
| Target Polysemy | Shadows | 影子 | 阴影 | synonyms |
| Target Polysemy | Quest | Quest | Avventura | synonyms |
| Source Polysemy | Worn | usé | vêtement | Gold: used, Predicted: cloth |
| Source Polysemy | Bitter | 苦 | 辛辣 | Gold: bitter (taste), predicted: bitter (feeling) |
| Antonyms | Unofficial | Ufficiale | Funzionario | funzionario: official |
| Antonyms | Mature | Mature | Jeune | Jeune: young |
| Antonyms | Afraid | Paura | Contento | Gold: fear, Predicted: happy |
| Common Words | Everybody | Jeder | Spaß | Gold: Everybody, Predicted: Fun |
| Common Words | Fourteen | Vierzehn | Dreirzehn | Numbers translated incorrectly |

(a) Sampled Errors

### 7.3 $\beta$ ORTHOGONALITY PROJECTION VS. AUTOENCODING LOSS

| Lang | Ortho | $\beta$ | | | Auto |
|---|---|---|---|---|---|
| | | 1e-2 | 1e-3 | 1e-4 | |
| en-de | 19.9 | 74.8 | 67.4 | 73.7 | 74.3 |
| en-ru | 102.5 | 40.8 | 30.7 | 36.7 | 46.1 |
| en-zh | 171.1 | 0 | 23.8 | 32.1 | 33.3 |

Table 6: Unsupervised accuracies for different values of $\beta$ (MUSE) and our autoencoding loss.

Lample et al. (2018) constraint the mapping matrix to be close to the manifold of orthogonal matrices by applying the following projection step after every update.

$$W \leftarrow (1 + \beta)W - \beta(WW^T)W$$

In our experiments we found out that the final accuracy is highly sensitive to the value of the hyper-parameter $\beta$ (Table 6). Our approach on the other hand uses an autoencoding loss which allows the model to flexibly adjusts the degree of orthogonality in a data driven manner and works consistently well for one choice of the scaling of the autoencoding loss.

## 7.4 HYPER-PARAMETERS

The following are the hyper parameters used in the experiments. The values separated by **/** are the different values tried in the parameter search.

- Number of words per language considered for GAN training: top 75000
- **Discriminator Parameters**:
    - embedding dim: 300
    - hidden dim: 2048
    - dropout prob: 0.1 (Only on the input layer)
    - label smoothing: 0.1
- **Generator Parameters**
- Initialization: Identity / Random Orthogonal
- Mean Center: True
- **GAN Training Parameters**
    - batch size: 32
    - Optimizer: SGD
    - Supervised loss optimizer: SGD / Adam
    - lr: 0.1 (with a schedule of 0.98 decay per round, and halved if unsupervised CSLS metric does not improve over two rounds).
    - Hubness Threshold: 20
- $f_a = cosine$

## 8 PERFORMANCE WITH DIFFERENT LEVELS OF SUPERVISION

| Model | en-es | es-en | en-fr | fr-en | en-de | de-en | en-ru | ru-en | en-zh | zh-en |
|---|---|---|---|---|---|---|---|---|---|---|
| Unsupervised | | | | | | | | | | |
| MUSE (U) | 81.7 | 83.3 | 82.3 | 82.1 | 74.0 | 72.2 | 44.0 | 59.1 | 32.5 | 31.4 |
| 50 Datapoints | | | | | | | | | | |
| MUSE (R) | 0.3 | 82.7 | 0.5 | 1.6 | 31.9 | 72.7 | 0.1 | 0.0 | 0.3 | 0.3 |
| GeoMM | 0.3 | 1.9 | 0.3 | 1.0 | 0.3 | 0.3 | 0.0 | 0.6 | 0.0 | 0.0 |
| RCSLS | 0.1 | 0.4 | 0.0 | 0.3 | 0.1 | 0.1 | 0.1 | 0.1 | 0.0 | 0.0 |
| BLISS (R) | 82.1 | 83.6 | 82.8 | 83 | 75.1 | 72.7 | 39.3 | 61 | 32.6 | 32.5 |
| 500 Datapoints | | | | | | | | | | |
| MUSE (R) | 81.6 | 83.5 | 82.1 | 82.0 | 73.1 | 72.7 | 40.3 | 62 | 34.5 | 32.2 |
| GeoMM | 31.9 | 46.6 | 34.4 | 44.7 | 13.5 | 14.7 | 10.6 | 20.5 | 3.9 | 2.9 |
| RCSLS | 22.9 | 44.9 | 22.4 | 43.5 | 9.9 | 10.2 | 7.9 | 19.6 | 6.6 | 7.1 |
| BLISS (R) | 82.3 | 83.4 | 82.3 | 82.9 | 74.7 | 73.1 | 41.6 | 63.0 | 36.3 | 35.1 |
| 5000 Datapoints | | | | | | | | | | |
| MUSE (R) | 81.9 | 82.8 | 82.2 | 82.1 | 75.2 | 72.4 | 50.4 | 63.7 | 39.2 | 36.3 |
| GeoMM | 79.7 | 82.7 | 79.9 | 83.2 | 71.7 | 70.6 | 49.7 | 65.5 | 43.7 | 40.1 |
| RCSLS | 80.9 | 82.9 | 80.4 | 82.5 | 72.5 | 70.9 | 51.3 | 63.8 | 42.5 | 41.9 |
| BLISS (R) | 82.4 | 84.9 | 82.6 | 83.9 | 75.7 | 72.5 | 52.1 | 65.2 | 42.5 | 42.8 |

Table 7: Performance with different levels of supervision.

Table 7 shows the performance of different models by varying the amount of supervised data points. We always outperform the unsupervised baseline method, as well as the supervised methods at the same level of supervision for most cases. Furthermore, we observe that iterative Procrustes refinement doesn't always yield the best model (measured in terms of the unsupervised CSLS metric). For low data points (50 and 500), we find that iterative Procrustes always helps. For 5000 datapoints, we find that it only helps when the languages have a low GH distance. Under an improved supervised loss function, BLISS (R) does not require iterative Procrustes refinement when all data is available.

