# OpenReview forum: "BLISS in Non-Isometric Embedding Spaces"
_ICLR.cc/2019/Conference_

### Official Review · AnonReviewer3 · 2018-10-30

**Rating:** 6
**Confidence:** 4

**Review:**

This paper presents a new semi-supervised method for bilingual dictionary induction and proposes a new metric to measure isometry between embedding spaces.

Pros:
- The paper proposes to use a new metric, the Gromov-Hausdorff distance to measure how isometric two word embedding spaces are.
- The toy example is useful for motivating the use case of the method.
- The approach achieves convincing results on the dataset.

Cons:
- Beyond the isometry metric, the main innovation as far as I can see seems to be the hubness filtering, which is incremental and not ablated, so it is not clear how much improvement it yields. The weak orthogonality constraint has already been used in [2].
- It is not clear to me what the proposed metric adds beyond the eigenvector similarity metric proposed in [1]. The authors should compare to this metric at least.
- The authors might want to add the results of [3] for an up-to-date comparison.

[1] Søgaard, A., Ruder, S., & Vulić, I. (2018). On the Limitations of Unsupervised Bilingual Dictionary Induction. In Proceedings of ACL 2018.
[2] Zhang, M., Liu, Y., Luan, H., & Sun, M. (2017). Adversarial Training for Unsupervised Bilingual Lexicon Induction. In Proceedings of ACL.
[3] Artetxe, M., Labaka, G., & Agirre, E. (2018). A robust self-learning method for fully unsupervised cross-lingual mappings of word embeddings. In Proceedings of ACL 2018.

---

> ### Author Response · Authors · 2018-11-26
> **Response to AnonReviewer3**
>
> Thank you for your feedback and insightful comments. Below we try and address your comments individually:
>
> >> Beyond the isometry metric, the main innovation as far as I can see seems to be the hubness filtering, which is incremental and not ablated, so it is not clear how much improvement it yields. The weak orthogonality constraint has already been used in [s 2].
>
> In addition to the Gromov-Hausdorff metric, our joint framework is a novel contribution. We show that it improves over both its corresponding supervised and unsupervised counterparts for two instantiations of our framework (BLISS(M) based on MUSE(S), and BLISS(R) based on RCSLS, incorporating reviewer feedback), which in turn illustrates its efficacy, with  BLISS(R) obtaining state-of-the-art results (to the best of our knowledge).
>
> >> It is not clear to me what the proposed metric adds beyond the eigenvector similarity metric proposed in [1]. The authors should compare to this metric at least.\
>
> Thank you for pointing this reference out; we have updated our paper to include the metric from [1] (Table 2). From the Table, we observe that our method correlates better than the eigenvector similarity metric.
>
> >> The authors might want to add the results of [3] for an up-to-date comparison.
>
> We have incorporated this baseline in our latest draft, along with an accompanying discussion (Section 4.2.4).

---

### Official Review · AnonReviewer1 · 2018-11-01
**Solid work with some obscure parts that should be clarified**

**Rating:** 6
**Confidence:** 5

**Review:**

This paper presents a new semi-supervised method to learn cross-lingual word embeddings mappings combining unsupervised distribution matching, alignment over a training dictionary, and a weak orthogonality constraint. The paper also shows that the underlying isometry assumption in orthogonal mappings weakens as the languages involved are more distant, proposes a new method to quantify the strength of the said assumption, and argues that the proposed semi-supervised mapping method is particularly suited for these more challenging cases.

I think that this is a solid work that explores an interesting research direction within cross-lingual embedding mappings. While the basic ingredients of the proposed method are not new, their combination is certainly original. In that regard, I think that the paper is rather incremental, but still has enough substance to make an interesting contribution. However, I think that some parts of the paper are too obscure, and I am not fully convinced by the experiments. I would appreciate if the authors could address my concerns below, and I would be happy to modify my score accordingly:

1) My understanding is that the proposed method (the one named BLISS in the experiments) only makes use of the proposed semi-supervised framework (Section 3.2) and is not followed by the iterative procrustes refinement (Section 3.3), but this is not clear at all from the paper. Could you clarify this?

2) It is well known that the retrieval method can have a big impact in bilingual dictionary induction due to the hubness problem. However, the paper does not detail which retrieval method is used in the experiments. I assume that MUSE uses CSLS and Vecmap uses nearest neighbor over cosine. Is this correct? What retrieval method does BLISS use?

3) I assume that when you talk about the "CSLS metric" in page 7 and 13 you refer to the unsupervised validation criterion of Lample et al. (Section 3.5 in their paper), and not to CSLS itself (Section 2.3 in their paper). In either case, this needs some clarification.

4) Unlike the "train" dictionaries, the "full" dictionaries from MUSE as provided at github also include the test set. Do you preprocess them to exclude the test set? If so, this should be clearly stated in the paper. If not, this would invalidate all these experiments.

5) The authors use different language pairs in their different result tables, which I find very confusing. For instance, none of the language pairs in table 5 (except for en-ru), are included in the main results (table 3), so we do not know how the different baselines and variants perform in them. Is there any reason for that?

6) Could you include all MUSE variants in Table 4?

7) While you compare your method to different versions of Vecmap (Artetxe et al., ACL 2017 & AAAI 2018), the last one (Artetxe et al., ACL 2018) (http://aclweb.org/anthology/P18-1073) is missing. That paper reports 48.1% and 48.2% accuracy for en-it and en-de in the unsupervised case, which is substantially better than your results for en-it (45.9%) and at par for en-de (48.3%). This goes against the main motivation of the paper (i.e. unsupervised distribution information and supervision from dictionaries can be combined for best results), as a completely unsupervised method seems to perform better than (or at least at par with) the proposed semi-supervised method. I think that the paper should include some discussion on this. In particular, I would like to know whether you have any argument to believe that both works are complementary.

---

> ### Author Response · Authors · 2018-11-26
> **Response to AnonReviewer1**
>
> Thank you for your feedback and insightful comments. Below we try and address your queries individually:
>
> 1) My understanding is that the proposed method (the one named BLISS in the experiments) only makes use of the proposed semi-supervised framework (Section 3.2) and is not followed by the iterative procrustes refinement (Section 3.3), but this is not clear at all from the paper. Could you clarify this?
>
> We always perform iterative Procrustes refinement. However, using iterative Procrustes refinement does not always yield the best performing model (where we measure performance using the unsupervised CSLS metric, and report the corresponding accuracy). We find that for very low data points, the supervision is noisy, and consequently, iterative Procrustes helps improve performance. However, when more data is available, iterative Procrustes only helps for languages that have low GH distance (see Table 2). Finally, under an improved supervised loss function and with sufficient data, BLISS (R) does not require iterative Procrustes refinement (Table 3 and 4). We have added this description in Appendix Section 8.
>
> 2) It is well known that the retrieval method can have a big impact in bilingual dictionary induction due to the hubness problem. However, the paper does not detail which retrieval method is used in the experiments. I assume that MUSE uses CSLS and Vecmap uses nearest neighbor over cosine. Is this correct? What retrieval method does BLISS use?
>
>
> BLISS uses CSLS; sorry for the confusion. We have clarified this in Section 4.2.2.
>
>
> 3) I assume that when you talk about the "CSLS metric" in page 7 and 13 you refer to the unsupervised validation criterion of Lample et al. (Section 3.5 in their paper), and not to CSLS itself (Section 2.3 in their paper). In either case, this needs some clarification.
>
>
> Thank you for pointing this ambiguity out, we meant the unsupervised validation criterion while referring to the CSLS metric. We have clarified this in the latest draft.
>
> 4) Unlike the "train" dictionaries, the "full" dictionaries from MUSE as provided at github also include the test set. Do you preprocess them to exclude the test set? If so, this should be clearly stated in the paper. If not, this would invalidate all these experiments.
>
>
> By “all”, we meant using all the data available in the training split (0-5000). However, since a word can translate into multiple words in the target language, 0-5000 effectively contains more than 5000 pairs. By “all”, we refer to using all the data points in the train split, whereas 5000 refers to using just the first 5000 pairs. In order to reduce ambiguity, we have removed the 5000 row from Table 3 in the latest draft. The performance with 5000 data points can be found in Table 5 (Table 6 in the original draft), where we show this information for a few language pairs due to space constraints, and in Appendix Section 8 for all language pairs.
>
> 5) The authors use different language pairs in their different result tables, which I find very confusing. For instance, none of the language pairs in table 5 (except for en-ru), are included in the main results (table 3), so we do not know how the different baselines and variants perform in them. Is there any reason for that?
> 6) Could you include all MUSE variants in Table 4?
>
> In the original draft, we had done this to accommodate for space constraints. However, based on reviewers’ feedback, we have added baseline numbers for both distant language pairs as well as on the VecMap dataset in the current version (Tables 2 and 4 respectively).
>
> 7) While you compare your method to different versions of Vecmap (Artetxe et al., ACL 2017 & AAAI 2018), the last one (Artetxe et al., ACL 2018) (http://aclweb.org/anthology/P18-1073) is missing. That paper reports 48.1% and 48.2% accuracy for en-it and en-de in the unsupervised case, which is substantially better than your results for en-it (45.9%) and at par for en-de (48.3%). This goes against the main motivation of the paper (i.e. unsupervised distribution information and supervision from dictionaries can be combined for best results), as a completely unsupervised method seems to perform better than (or at least at par with) the proposed semi-supervised method. I think that the paper should include some discussion on this. In particular, I would like to know whether you have any argument to believe that both works are complementary.
>
> Artexe et al. (2018) translate in a common embedding space, while our method, similar to RCSLS, translate in the target embedding space. It was shown in  Kementchedjhieva et al. (2018) that translating in a common embedding space leads to performance gains, which we believe to be the case here. We have added the numbers of Artexe et al. (2018) as well as GeoMM (a supervised method which translates in the common embedding space) in Table 4, and also included a discussion stating the same (Section 4.2.4).

---

### Official Review · AnonReviewer2 · 2018-11-03
**a semi-supervised algorithm for bilingual lexicon induction problem**

**Rating:** 4
**Confidence:** 5

**Review:**

Summary:

The paper propose a semi-supervised algorithm for bilingual lexicon induction (BLI) problem. Prior works on BLI problem usually impose orthogonality constraint on the linear transformation in order to obtain a "reversible" mapping and to preserve the monolingual performance. However, from both modeling and generalization perspective, recent works do not impose this constraint while learning the mapping (Doval et al 2018, Jawanpuria et al 2018, Joulin et al 2018, Sogaard et al 2018, among others). The present work argues for the removal of the orthogonality constraint when language spaces are non-isometric, and proposes to employ the Gromov Hausdroff (GH) distance to validate this condition. Overall, the paper employs  an objective function which is the sum of the (unsupervised) adversarial distribution matching objective (Lample et al 2018b), (supervised) the BLI loss function (typically the square loss), and a consistency loss (Hoshen and Wolf 2018). Empirically, the proposed method shows better results than unsupervised method of Lample et al (2018b) and the Procrustes solution.

The pros:

- Existing works have shown that some BLI techniques perform better than the other in *some* pair of languages. Hence, it seems that there may not be "one size fit all" BLI technique. The proposed usage of GH distance is in the direction to quantitatively categorize pairs of languages. Based on a carefully crafted metric, the practical systems may chose to use one BLI algorithm over another for a given pair of languages.

The cons:

- From modeling perspective, the utility of weak orthogonality constraint in the objective function is unclear. Does it improve generalization performance? Is it for preserving monolingual performance? The cited works (in the above summary) show that removing the strong/weak orthogonality constraint improves the BLI accuracy while preserving the monolingual performance.
- The baselines chosen for experiments are not state-of-the-art. In addition, Artetxe et al. (2017, 2018) results are with NN/ISF retrieval procedure. These baselines should be rerun with CSLS retrieval procedure (codes are available in the author's website), which is now a standard for BLI task. Refer to Artetxe et al (2018b), Joulin et al (2018), Jawanpuria et al (2018), Gravel et al (2018) for state-of-the-art (semi-supervised/ unsupervised) results on MUSE and Vecmap datasets.
- Experiments with varying data (Table 6) does not provide a clear picture without discussing unsupervised/semi-supervised baselines.
- The logic behind experiments on GH distance (Table 2) is unclear. Why should a high correlation with *a baseline* suggest that GH distance correlates well with the degree of isometry of the two languages? Does GH distance has high correlation with *any* baseline for BLI?


Artetxe et al (2018b): A robust self-learning method for fully unsupervised cross-lingual mappings of word embeddings.
Joulin et al (2018): Loss in translation: Learning bilingual word mapping with a retrieval criterion.
Jawanpuria et al (2018): Learning multilingual word embeddings in latent metric space: a geometric approach.
Hoshen and Wolf (2018): Non-adversarial unsupervised word translation.
Doval et al (2018): Improving cross-lingual word embeddings by meeting in the middle.

---

> ### Author Response · Authors · 2018-11-26
> **Response to AnonReviewer2**
>
> Thank you for your feedback and comments. Below, we try and address your comments individually:
>
> >> From modeling perspective, the utility of weak orthogonality constraint in the objective function is unclear. Does it improve generalization performance? Is it for preserving monolingual performance? The cited works (in the above summary) show that removing the strong/weak orthogonality constraint improves the BLI accuracy while preserving the monolingual performance.
>
> Our weak orthogonality constraint stabilizes training and reduces reliance on hyperparameters. In particular, as described in Section 7.3 and Table 6 in the appendix, we observe that the performance of MUSE is very sensitive to the choice of the hyperparameter Beta used in their orthogonal projection step . On the other hand, the data-driven weak orthogonality constraint is parameter independant and is more robust than its data-independant counterpart. Hence it is invariant of the language pair, thereby generalizing better.
>
> Compared to this framework, Jawanpuria et al. (2018) learn orthogonal mapping into a common embedding space. This framework of learning orthogonal mapping was also adopted by Søgaard et al. (2018) who do iterative refinement over a small seed dictionary containing identical seed words (similar in spirit to our MUSE(R) baseline).
>
>  In the initial work of Joulin et al. (2018), they apply a data independent orthogonality constraint, and constrain the weight matrices to have eigenvalues <= 1 (spectral norming), or by constraining the matrices to be unit Frobenius norm. In their final published work, they remove these constraints altogether, and show improvements in performance. However, this final work was concurrent, being published less than a month prior to the ICLR deadline.
>
> >> The baselines chosen for experiments are not state-of-the-art. In addition, Artetxe et al. (2017, 2018) results are with NN/ISF retrieval procedure. These baselines should be rerun with CSLS retrieval procedure (codes are available in the author's website), which is now a standard for BLI task. Refer to Artetxe et al (2018b), Joulin et al (2018), Jawanpuria et al (2018), Gravel et al (2018) for state-of-the-art (semi-supervised/ unsupervised) results on MUSE and Vecmap datasets.
>
> We apologise for missing these baselines. Our submission posited a semi-supervised framework, and we had compared against the corresponding supervised and unsupervised counterparts.  We have updated our paper to include more sophisticated supervised baselines (namely, Joulin et al. (2018) and Jawanpuria et al. (2018)). For fair comparison, we include an instantiation of our semi-supervised framework with the supervised CSLS loss of Joulin et al. (2018), and show that this still outperforms its supervised and unsupervised counterparts.
>
> >> Experiments with varying data (Table 6) does not provide a clear picture without discussing unsupervised/semi-supervised baselines.
>
> We apologize for being brief in the description of Table 6 (Table 5 in the updated version). Note that BLISS consistently outperforms its unsupervised counterpart MUSE(U), even with minimal amounts of data, as well as MUSE(R), which is a strong semi-supervised  method, utilizing dictionary expansion via iterative Procrustes refinement. Based on your feedback, we also incorporate a comparison with RCSLS (which uses CSLS, and is a semi-supervised method). We added clarifying details explaining this in Section 4.2.1, as well as a detailed comparison of the performances of the different unsupervised and semi-supervised methods across language pairs in Section 8 in the Appendix
>
> >> The logic behind experiments on GH distance (Table 2) is unclear. Why should a high correlation with *a baseline* suggest that GH distance correlates well with the degree of isometry of the two languages? Does GH distance has high correlation with *any* baseline for BLI?
>
> In order to compute the orthogonality of spaces empirically, we wanted to avoid relying on matched lexicons, and consequently chose an unsupervised method (MUSE(U)). In addition, incorporating comments from Reviewer 3, we also show that our method correlates with accuracies better than Søgaard’s method across several baseline techniques for BLI.
>
> References
> Jawanpuria et al. (2018): Learning multilingual word embeddings in latent metric space: a geometric approach.
> Søgaard et al. (2018): On the Limitations of Unsupervised Bilingual Dictionary Induction.
> Joulin et al. (2018): Loss in translation: Learning bilingual word mapping with a retrieval criterion.

---

### Author Response · Authors · 2018-11-26
**Common Response to all reviewers**

We thank the reviewers for their detailed feedback.

The general feedback received from all reviewers suggested adding baselines that we had originally missed and comparing our framework against them, as well as adding clarifying information about details we missed in our original submission. Based on the feedback received, we have updated our submission. Our changes can be summarized as follows:

1. We added a baseline (Søgaard et. al.) for measuring the degree of isometry between two embedding spaces, and compare its correlation with accuracies (as done in Søgaard et. al.) against our proposed GH distance-based metric.
2. We included SoTA methods (Joulin et al. (2018), Jawanpuria et al. (2018), Artetxe et al. (2018)), as pointed out by the reviewers, and show that they fit in nicely into our proposed semi-supervised framework. We also update our tables to compare against these supervised methods (Table 3 and Table 4).
3. We also added the missing baseline accuracies for etymologically distant language pairs (Table 2).
4. In addition, we restructured and added clarifying information for improved clarity based on reviewer feedback.

We would again like to thank the reviewers for their positive and helpful insights.

References
Søgaard et al. (2018): On the Limitations of Unsupervised Bilingual Dictionary Induction.
Joulin et al. (2018): Loss in translation: Learning bilingual word mapping with a retrieval criterion.
Jawanpuria et al. (2018): Learning multilingual word embeddings in latent metric space: a geometric approach.
Artetxe et al. (2018): A robust self-learning method for fully unsupervised cross-lingual mappings of word embeddings.

---

> ### Public Comment · (anonymous) · 2018-11-29
> **The performance gain of BLISS(R) mainly comes from the optimization towards CSLS metric?**
>
> I saw in the updated version, the performance of BLISS (R) matches the sota results. However, I was wondering if this gain mainly comes from the optimization towards CSLS metric in your supervised loss, which is the main contribution of Joulin et al. (2018)' method.

---

> > ### Author Response · Authors · 2018-11-30
> > **Response to question**
> >
> > Thank you for your question.
> >
> > Although optimizing over the CSLS loss (BLISS(R)) improves over optimizing over the cosine loss (BLISS(M)), we show that both these instantiations of our proposed semi-supervised framework outperform their supervised and unsupervised counterparts. The SotA results of BLISS(R) thus come from the proposed semi-supervised framework in combination with RCSLS 's (Joulin et al. (2018)) supervised objective, as opposed to from purely the CSLS objective.
> >
> > This is especially visible from Table 2, BLISS(R) substantially outperforms RCSLS on 6 of 9 language pairs, especially when the GH distance between the languages is high (2.4% for en-uk, 3.4% for en-sv, 0.9% for en-el, 0.8% for en-hi, 2.4% for en-ko). Tables 3 and 4 underscore this point, wherein the model performs at least at par with (and often better than) RCSLS on European languages, and performs significantly better on en-zh (2.8%) and zh-en (0.9%); while on the VecMap dataset, an improvement of 0.8% is observed for both languages.
> >
> > Similarly, BLISS(M) outperforms its supervised counterpart MUSE(S). This can be seen from Tables 3 and 4, where BLISS(M) outperforms MUSE(S) in 10 of 12 language pairs.
> >
> > In addition, as highlighted by Tables 5 and 7, the semi-supervised framework is able to function even when the model has low levels of supervision, a case where RCSLS does not perform well at all, achieving very low accuracies for 50 and 500 data-points.
> >
> > To summarize, the semi-supervised framework improves over both its supervised and unsupervised counterparts, and does extremely well when the available training data is less.

---

### Meta-Review · Area_Chair1 · 2018-12-02
**Reject**

**Confidence:** 4
**Recommendation:** Reject

**Metareview:**

This paper is very close to the decision boundary and the reviewers were split about whether it should be accepted or not. The authors updated the paper with additional experiments as request by the reviewers.
The area chair acknowledges that there is some novelty that leads to (moderate) empirical gains but does not see these as sufficient to push the paper over the very competitive acceptance threshold.